# Exploring the colloid-to-polymer transition for ultra-low crosslinked microgels from three to two dimensions

A. Scotti [1], S. Bochenek[1], M. Brugnoni [1], M.A. Fernandez-Rodriguez[2], M.F. Schulte [1], J.E. Houston [3,4], A.P.H. Gelissen[1], I.I. Potemkin[5,6,7], L. Isa [2] & W. Richtering [1,8]

Microgels are solvent-swollen nano- and microparticles that show prevalent colloidal-like behavior despite their polymeric nature. Here we study ultra-low crosslinked poly(N-iso-propylacrylamide) microgels (ULC), which can behave like colloids or flexible polymers depending on dimensionality, compression or other external stimuli. Small-angle neutron scattering shows that the structure of the ULC microgels in bulk aqueous solution is characterized by a density profile that decays smoothly from the center to a fuzzy surface. Their phase behavior and rheological properties are those of soft colloids. However, when these microgels are confined at an oil-water interface, their behavior resembles that of flexible macromolecules. Once monolayers of ultra-low crosslinked microgels are compressed, deposited on solid substrate and studied with atomic-force microscopy, a concentration-dependent topography is observed. Depending on the compression, these microgels can behave as flexible polymers, covering the substrate with a uniform film, or as colloidal microgels leading to a monolayer of particles.

[1] Institute of Physical Chemistry, RWTH Aachen University, 52056 Aachen, Germany. [2] Laboratory for Interfaces, Soft Matter and Assembly, Department of Materials, ETH Zurich, 8093 Zurich, Switzerland. [3] Jülich Centre for Neutron Science (JCNS) at Heinz Maier-Leibnitz Zentrum (MLZ) Forschungszentrum Jülich GmbH, 85748 Garching, Germany. [4] European Spallation Source ERIC, Box 176, SE-221 00 Lund, Sweden. [5] Physics Department, Lomonosov Moscow State University, 119991 Moscow, Russian Federation. [6] DWI - Leibniz Institute for Interactive Materials, Aachen 52056, Germany. [7] National Research South Ural State University, Chelyabinsk 454080, Russian Federation. [8] JARA-SOFT, 52056 Aachen, Germany. Correspondence and requests for materials should be addressed to A.S. (email: andrea.scotti@rwth-aachen.de) or to W.R. (email: richtering@rwth-aachen.de)

Aqueous solutions of flexible polymers and suspensions of colloids are two of the most important classes of materials in soft matter. Flexible polymers, which bend at length scales longer than their Kuhn length, have high degrees of entanglement and interpenetration, typical for linear polymers[1]. In contrast, colloids maintain their individuality as particles and show phase transitions between disordered and crystalline arrangements in both two and three dimensions[2].

Microgels are crosslinked polymers swollen by the solvent. Typically they have a spherical shape with sizes in the range of nm to μm. These nano- and microparticles, constituted of polymeric networks, have properties of both flexible polymers and colloids. As a polymer, their swelling is affected by the solvent quality, and microgels can present volume phase transitions, i.e., a microgel-to-colloid transition, depending on external stimuli[3,4]. Furthermore, in the swollen state they are soft and can be compressed or interpenetrate once in concentrated suspensions[5,6]. Nevertheless, their flow properties and phase behavior are the same as those of colloids interacting via soft potentials. This is due to the less crosslinked fuzzy periphery of microgels, which is composed of dangling polymeric chains. In addition, microgels are highly interfacially active and can be employed, similarly to colloids, as emulsifiers[7,8].

Due to the predominant colloidal behavior, poly(N-isopropylacrylamide)-based (pNIPAM) microgels have been widely used as a model system for soft spheres to investigate fundamental questions related to strong and fragile glass formers[9,10], to melting–freezing[11–13] and to solid–solid phase transitions, both in two (2D)[14–16] and three dimensions (3D)[17,18].

In this study, we address the microgel-to-polymer transition and ask the question of whether microgels can be prepared with a predominant flexible polymer nature.

Obviously, the amount of crosslinker within the polymer network is responsible for the individual shape and swelling of the microgels. A higher incorporation of crosslinker produces stiffer microgels that behave more similar to hard incompressible colloids[11]. In contrast, therefore, decreasing the amount of crosslinker as much as possible will lead to microgels where the colloidal nature should be minimized. Here, we use ultra-low crosslinked pNIPAM microgels. They represent the softest microgels that can be obtained by precipitation polymerization. Although no crosslinking agents are present during the synthesis, the formation of a polymeric network is still promoted via transfer reactions, leading to extremely soft micro- and nanogels[19–21].

We investigate ultra-low crosslinked microgels in bulk aqueous solution as well as at fluid interfaces. Our results show that while the ultra-low crosslinked (ULC) microgels present a colloidal-like behavior in three dimensions, their nature changes once they are confined at interfaces. In two dimensions, the interplay between the extreme softness of the polymer network and the action of surface tension leads the ULC microgels to show properties typical of flexible polymers within a certain concentration range, as it can be seen in their compression isotherms or by the observed microstructure of deposited monolayers.

## Results

**Small-angle neutron scattering.** For regular microgels, the addition of crosslinker, e.g., N,N′-methylenebisacrylamide (BIS), has two important consequences: first, it enforces the stiffness of the network; second, it leads to a characteristic fuzzy sphere architecture. This is due to the fact that the crosslinker has a faster reaction rate than the monomer leading to the formation of a more crosslinked core with higher polymer density, surrounded by a fuzzy corona with less crosslinks and a lower polymer

density[22,23]. As a consequence of this architecture, the scattered intensity measured in a small-angle neutron (or X-ray) scattering (SANS) experiment of diluted suspensions of microgels presents typical oscillations. This profile can be fitted using a model that accounts for a denser core with radius $R_c$, surrounded by a fuzzy shell of length $2\sigma_s$, with decreasing polymer content[5,22]. The smaller is the core, the more inhomogeneous is the polymer distribution within the microgel. Their form factor is similar to that of hard colloids, except for the less defined periphery. In contrast, it is significantly different with respect to the form factor of flexible polymers that does not show any oscillation, but after a plateau for low scattering vector, $q$, has a monotonic decrease[24,25].

Figure 1a shows the SANS intensity plotted versus $q$ obtained by measuring a diluted suspension of ULC microgels in heavy water below and above their volume phase transition temperature (VPTT), at $T = 20.0 \pm 0.5$ and $40.0 \pm 0.5\,°C$, respectively. The total size of the microgel is $R = R_c + 2\sigma_s$. The structure

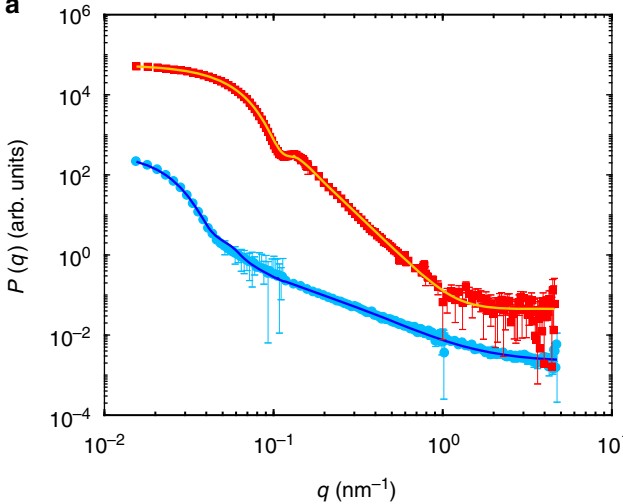

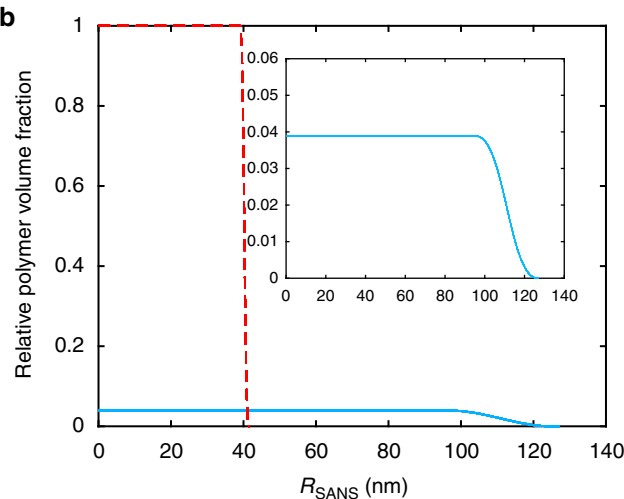

**Fig. 1** Small-angle neutron scattering data and analysis. **a** Small-angle neutron scattering form factor, $P(q)$, versus scattering vector, $q$, for ultra-low crosslinked (ULC) microgels in dilute suspensions of $D_2O$ at $20.0 \pm 0.5\,°C$, light blue points, and $40.0 \pm 0.5\,°C$, red squares. The solid lines are fits with the fuzzy sphere model[22]. **b** Relative polymer volume fraction versus radius obtained from the data fit for ULC microgels at $20.0 \pm 0.5\,°C$, light blue solid line, and $40.0 \pm 0.5\,°C$, red dashed line. Inset: zoom of the relative polymer volume fraction versus radius for the ULC microgels below the volume phase transition temperature (VPTT)

parameters are obtained from the fits, and the radial distributions are shown in Fig. 1b.

The microgels have a total radius in the swollen state (light blue circles) of $126 \pm 1$ nm with $R_c = 96.1 \pm 0.5$ nm and $2\sigma_s = 29.8 \pm 0.3$ nm. The fuzziness of the shell of the ULC microgels is less pronounced as compared to regularly crosslinked microgels[6,13,22,26], but still the ULC microgels have the typical fuzzy sphere architecture. Thus, the polymer distribution within ULC microgels is more homogeneous than for microgels synthesized with additional crosslinker. The average mesh size of the polymer network is estimated to be $24 \pm 1$ nm, which is significantly larger than for regular crosslinked microgels (Supplementary Note 5)[22]. This means that even if ULCs have a more homogeneous crosslinker distribution, the absolute number of crosslinks is lower compared to regular crosslinked microgels. The polydispersity is $11.3 \pm 0.7\%$. The ULC microgels collapse to $41 \pm 1$ nm and the fit shows that the external fuzziness disappears once the temperature is above the VPTT of pNIPAM. For collapsed ULC microgels, the fuzzy sphere model produces the same results as a fit performed with the simple hard sphere model (Supplementary Note 4).

Sizes and polydispersities obtained by SANS agree with the hydrodynamic radii obtained below and above the VPTT by the analysis of dynamic light scattering (DLS) data shown in the Supplementary Note 2, $134 \pm 1$ nm and $44.8 \pm 0.2$ nm, respectively. Estimating the amount of crosslinks within the polymeric network is difficult. A possible method is to compare the swelling ratios of the microgels synthesized with and without crosslinker agent[27]. This comparison shows that the ULC microgels contain less crosslinks than microgels with comparable size and synthesized with low amounts of crosslinker agent (Supplementary Note 3).

**Bulk phase behavior.** A marked difference between flexible polymers and colloidal suspensions is that the former show a pronounced entanglement and interpenetration with increasing concentrations. In contrast, colloids reveal a liquid-to-solid transition depending on the concentration, which is connected to a transition from a disordered fluid to a colloidal crystal with a coexistence region in between liquid and fully crystalline samples. Similarly, microgels can form crystals but at higher concentrations with respect to rigid colloids due to their soft interparticle potential[11,13]. A shift of the transition boundaries is observed for other colloids interacting with soft potentials, e.g., slightly charged hard spheres[28].

Bulk suspensions of the ULC microgels were investigated to understand if their extreme softness makes their phase behavior different with respect to that of the colloids. The phase behavior of microgels is a function of the generalized volume fraction, $\zeta = (NV_{\text{Swollen}})/V_{\text{tot}}$ where $N$ is the number of microgels in the sample and $V_{\text{Swollen}}$ and $V_{\text{tot}}$ are the volume of the microgel in the swollen state and the total volume of the sample, respectively. In case of rigid colloids, $\zeta$ equals the volume fraction $\phi$; however, for soft microgels $\zeta$ can reach values well above 0.74 reflecting deformation[29], deswelling[5,13] or interpenetration[6]. The generalized volume fraction, $\zeta$, is proportional to the mass concentration of polymer within the suspension, $c$, via a conversion constant, $k$: $\zeta = kc$. $k$ is obtained using viscosimetry[11] as shown in the Supplementary Note 6.

A series of samples, covering a $\zeta$ range between $0.50 \pm 0.01$ and $3.02 \pm 0.07$, was made and stored at $T = 20.0 \pm 0.5$ °C. The freezing point, i.e., the onset of the coexistence of liquid and crystals, was found at $\zeta_f = 0.80 \pm 0.02$. With increasing $\zeta$, fully crystalline samples were observed above $\zeta_m = 0.83 \pm 0.02$; this value is called the melting point. Before the samples became a

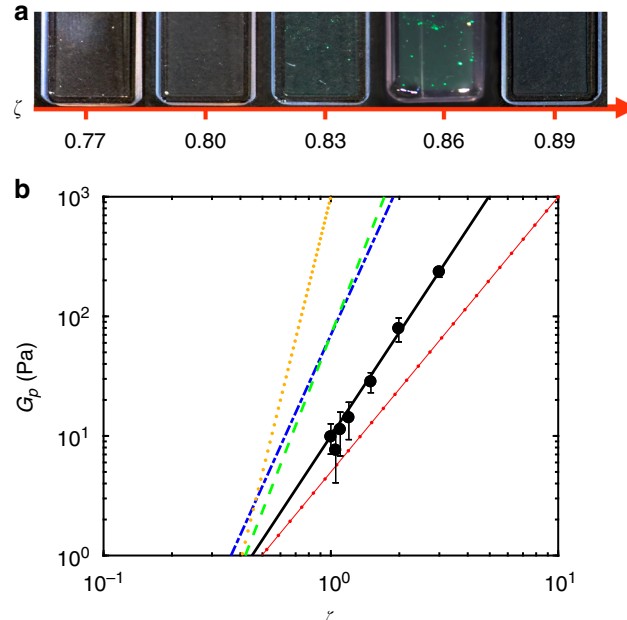

**Fig. 2** Phase behavior and $G'$ plateau of suspensions of ultra-low crosslinked (ULC) microgels. **a** Image of a series of samples of ultra-low crosslinked microgels of increasing volume fraction in water. **b** Plateau value of $G'$, $G_p$ (black circles), versus the generalized volume fraction, $\zeta$. Black line: data fit with $G_p \propto \zeta^m$. Dotted, dashed, dotted-dashed and line-with-dots lines are taken from refs. [1,11,32,33], see text for details

glass, at $\zeta = 0.89 \pm 0.02$, crystalline samples are reported in Fig. 2a, demonstrating that ULC microgels behave like other colloids interacting with soft potentials[11,13,28,30].

The softness is solely shifting the boundaries of the phase transitions to higher $\zeta$ with respect to regularly crosslinked microgels[11,13]. This is a consequence of concomitant effects from deswelling[13], interpenetrating[6] and the interparticle potential[31]. Indeed, the potential between microgels can be modeled as $\Phi(r) \propto r^{-n}$, where $r$ is the microgel–microgel distance and $n$ contains information on the softness; smaller values of $n$ correspond to softer potentials. The form of $\Phi(r)$ leads to a power-law dependence of the plateau modulus of $G'$ on the volume fraction: $G_p \propto \zeta^m$ [32]. The black line in Fig. 2b represents a fit of the data to $\zeta^m$ that leads to $m = 2.88 \pm 0.02$. This exponent is linked to the exponent $n$ by the equation: $m = 1 + n/3$. The $n = 5.6 \pm 0.1$ is estimated from rheology measurements as a function of $\zeta$ (Supplementary Note 7). This value is significantly lower than for regular crosslinked microgels where it lies between 9 and 20[11,13,32,33]. Nevertheless, ULC microgels still behave as colloids; for flexible macromolecules a linear behavior with a slope $n = 2.3$ is expected (red line with dots in Fig. 2b)[1].

**Mechanical response of monolayers.** Next we present the behavior of ULC microgels under two-dimensional confinement. For this purpose, the microgels are placed at the oil–water interface in a Langmuir–Blodgett trough. This allows us to determine the surface pressure as a function of concentration and simultaneously deposit the microgel monolayers onto a solid substrate. The structure of the deposited films of microgels is then probed via atomic force microscopy (AFM) in the dry state on the solid substrate.

In Fig. 3 the compression isotherms of ultra-low crosslinked microgels (black lines) are presented. The surface pressure, $\pi$, is plotted against the area normalized by the added mass of microgel (Area/Mass). For comparison we also report isotherms

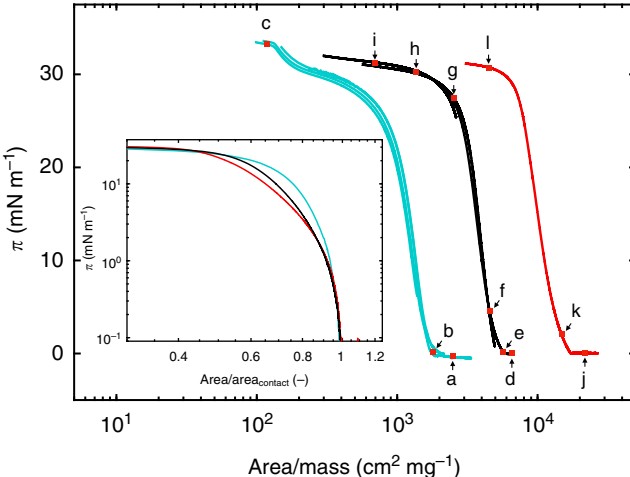

**Fig. 3** Compression isotherms for linear, ultra-low crosslinked (ULC) and regular crosslinked microgels. Surface pressure, $\pi$, versus the area normalized by added mass of microgels (Area/Mass) for 5 mol% crosslinked microgels (cyan lines), ULC microgels (black lines) and linear poly($N$-isopropylacrylamide) (pNIPAM; red line) at $T = 20.0 \pm 0.5$ °C. The red squares with letters a–l indicate the $\pi$ values of the atomic force micrographs in Fig. 4a–l. Inset: Plot of $\pi$ versus the trough area normalized by the trough area at the contact point, Area/Area$_{contact}$, of 5 mol% crosslinked microgels (cyan line), ULC microgels (black line) and linear pNIPAM (red line) at $T = 20.0 \pm 0.5$ °C in log–log scale

of regularly crosslinked microgels (containing 5 mol% crosslinker, cyan lines, $R_h = 153.2 \pm 0.6$ nm at $T = 20.0 \pm 0.5$ °C) and linear pNIPAM (red line, $M_w \approx 10^2$ kg mol$^{-1}$).

The cyan curves in Fig. 3 exhibit two distinct increases after the contact. The first is between Area/Mass $\approx 6 \times 10^2$ cm$^2$ mg$^{-1}$ and $\approx 2 \times 10^3$ cm$^2$ mg$^{-1}$. The second is between $\approx 1.4 \times 10^2$ cm$^2$ mg$^{-1}$ and $\approx 2.5 \times 10^2$ cm$^2$ mg$^{-1}$. This two-stepped course is typical for microgels synthesized with the addition of crosslinker[14,34–36] and is due to their well-defined core–shell structure. Upon adsorption, microgels are deformed, but their well-defined crosslinking density profile produces heterogeneities in polymer fraction laterally[37] and vertically[38] to the interface. The different stiffness of the core and the shell under compression is the reason for the presence of the two distinct plateaus.

The linear homopolymers (Fig. 3 red curve) have a single-stepped course, in agreement with literature[39]. The reason for this is that flexible polymers can fully spread and cover the interface uniformly[40]. Surprisingly, the compression isotherms of ultra-low crosslinked microgels (Fig. 3, black lines) exhibit a single-step increase, virtually identical to the linear pNIPAM. This also indicates that the ULC microgels cover the interface homogeneously due to the absence of a well-defined crosslinking density profile as highlighted by SANS. In other words, they behave as flexible polymers once confined at the interface. In contrast, for an ideal gas of hard spheres, $\pi$ is expected to be determined by the excluded area effects and the collision rate before diverging at the maximum packing fraction in 2D $(\pi/(2\sqrt{3}))$[41].

The inset of Fig. 3 gives a closer look at the course of the compression isotherms. Here it is clear that the differences between the linear polymer and the ULC microgels (red and black lines) are less pronounced than the differences between ULC microgels and regularly crosslinked microgels (black and cyan lines). Nevertheless, deviations between the ULC microgels and the linear pNIPAM are visible for $2$ mN m$^{-1} \lesssim \pi \lesssim 25$ mN m$^{-1}$. Those differences are larger than the experimental errors associated to the repetition of the measurements (Supplementary

Note 8). This suggests that the monolayer of the ULC microgels presents characteristic features. We further investigate this by determining the structure of the monolayers of all three systems. We have previously demonstrated that a Langmuir–Blodgett-type deposition of the microgels from the oil–water interface onto a solid substrate does not affect the structural order of the microgels in the monolayer[42]. Consequently, the phase behavior of microgels at interfaces can be investigated after deposition on a solid substrate by means of AFM in dry state[14].

**Two-dimensional phase behavior**. AFM micrographs are reported in Fig. 4 for regular 5 mol% crosslinked microgels (cyan box), ULC microgels (black box) and linear polymers (red box). Regular crosslinked microgels present a pronounced core–shell structure at the interface (Fig. 4a). In literature, it is shown that this fried-egg structure is observed even when a very low amount of crosslinker is added during the synthesis[43]. This indicates that microgels synthesized with little addition of crosslinker always present a more crosslinked core, harder to deform, surrounded by a soft fuzzy shell.

In contrast to regular microgels, before contact ULC microgels are strongly spread and flattened at the substrate into disks. The ULC microgels have a uniform height of $\approx 1$ nm, surrounded by a very thin, less dense fuzzy shell, as shown in Fig. 4d. The ratio between the radii in two and three dimensions for ULC microgels is 2.3, while for the regularly crosslinked microgels in Fig. 4a is 1.9. This ratio is a measure of the microgel stretching. The larger ratio of the ULC microgels suggests that their polymeric network is more stretched at the interface with respect to regularly crosslinked microgels. This observation and the fact that ULC microgels have a larger mesh size in bulk suggest that even at the interface the mesh size of ULC microgels is larger.

It is important to notice that ULC microgels maintain an individual and almost circular shape, while linear polymers are not distinguishable from the substrate (Fig. 4j). The higher capability of a flexible polymer to spread produces an extremely flat profile[40]. The individual polymers are indistinguishable from the substrate due to a height that is comparable to the substrate roughness and the finite dimensions of the tip[44].

When the monolayers are compressed, the regular 5 mol% crosslinked microgels remain clearly distinguishable and arrange in a hexagonally packed lattice (Fig. 4b) as expected for colloids. Figure 4e, however, shows that ULC microgels form a uniform polymer film where the single microgels are indistinguishable in both the phase and the height images, Supplementary Fig. 8. This behavior is typical for both flexible polymers where, under compression, interpenetration and entanglement are not restricted by crosslinks and soft macromolecules, e.g., arborescent microgels at liquid interfaces[45], where strong deformations produce a uniform coverage of the interface. Thus, at low compressions, ULC microgels cover uniformly the interface in contrast to colloids that always maintain their individual shape. Whether this is due to strong interpenetration, large deformation or both cannot be deduced from Fig. 4e.

Upon further compression an increase of the monolayer roughness is registered (Supplementary Fig. 9) and, as a consequence, the ULC microgels become clearly distinguishable again. The evolution shown in Fig. 4f–i makes ULC microgels different to flexible polymers confined at the interface where individual polymers remain indistinguishable independent of the compression (Fig. 4k, l).

Similar to the three-dimensional case, a significant difference between spherical colloids and flexible polymers at interfaces is that the former can have three different phases: liquid, hexatic and crystalline. These phases are characterized by two

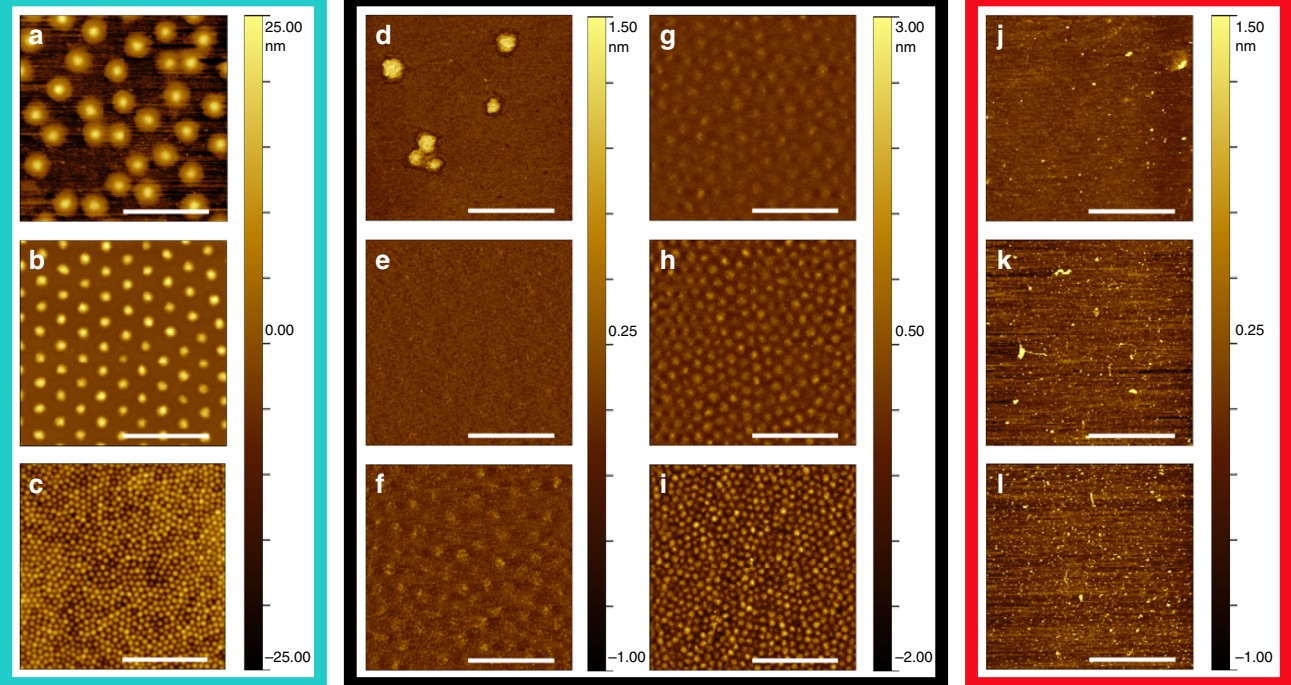

**Fig. 4** Atomic force microscopy (AFM) micrographs of deposited monolayers on solid substrates. Series of AFM images in dry state corresponding to the positions **a**–**l** in Fig. 3 for 5 mol% crosslinked microgels (cyan box), ultra-low crosslinked (ULC) microgels (black box) and linear poly(*N*-isopropylacrylamide) (pNIPAM; red box) after deposition of the monolayer to a solid substrate at $T = 20.0 \pm 0.5\,°C$. Scale bars correspond to 2 μm

second-order transitions, meaning that no fluid-crystal or hexatic-crystal regimes can coexist[2]. The regular microgels in Fig. 4b show a hexagonal lattice that under compression evolves into another hexagonal lattice with a smaller lattice constant (Fig. 4c). The evolution from a crystal where microgels are in shell–shell contact to a new lattice characterized by a core–core contact is well illustrated in the literature[14,36]. For our regular microgels this is confirmed by both the radial distribution functions corresponding to 4b, c and by analysis of the AFM micrograph during the transition in Supplementary Fig. 13.

Figure 4f–i highlights that no crystallization of the ULC microgels at the interface is observed. The reason for the suppression of crystallization is the increase of size polydispersity of the ULC microgels once confined in 2D that passes from 11.3 ± 0.7% in bulk to 23 ± 5% (Supplementary Fig. 12). This can either significantly slow down the crystallization kinetic as observed in polydisperse suspensions of microgels in 3D[13] or suppress the crystallization since polydispersity has a value higher than the theoretical limit that hinders crystallization for hard disks in two dimensions[46].

This significant change between the size distribution in bulk and at the interface can be understood by considering that microgels radially stretch and deform upon adsorption[36,47] due to the interfacial free energy reduction as far as the internal elasticity of the network allows it. SANS and DLS both show that ULC microgels have a softer network and a less defined crosslinking density profile, i.e., they are more homogeneous, compared to regularly crosslinked microgels. Larger variations in the topology of the networks of different ULC microgels can be expected, causing significant size variations upon interfacial adsorption.

## Discussion
To summarize, in this study we have shown that ultra-low crosslinked microgels behave as colloids in three dimensions but not in two, where within a certain range of concentrations their

properties are very similar to those of flexible polymers. Therefore, a key parameter for the predominance of the colloidal behavior is represented by the interplay between the softness of the network and the dimensionality.

In three dimensions, SANS highlights that ULC microgels are spherically shaped with a fuzzy shell architecture typical of microgels[22]. However, they present a large homogeneous core surrounded by a thin fuzzy shell. Furthermore, their phase behavior is the same as that of colloids with liquid, crystalline and glassy phases[3,28,30]. Rheology measurements show that their interaction potential is softer than for usual microgels but still different from the scaling law of flexible polymers[1,11,31]. The sole consequence of the soft interaction potential is to shift the onset of the liquid-to-crystal transition to higher concentrations as expected for soft microgels[11,13].

The confinement of ULC microgels at oil–water interfaces makes their polymeric nature dominant over their colloidal behavior. The compression isotherms of ULC microgels show a single-step increase as expected for flexible homopolymers[39]. Their weakly crosslinked network spreads more than regular crosslinked microgels creating a homogeneous coverage similar to linear pNIPAM. Nevertheless, a unique feature of ultra-low crosslinked microgels with respect to other soft objects is that with increasing concentration the topography at the interface changes.

This can be explained considering that the shape of regularly crosslinked microgels is controlled by a competition between the elasticity of the network, which causes a spherical shape, and the gain in the interfacial energy due to adsorption of polymer subchains. The maximum gain would be achieved if all subchains are adsorbed (pancake-like shape), but this would be accompanied by a high penalty in elasticity. As a result, regular microgels at the solid interface show the typical fried-egg structure[43].

In case of ULC microgels, the amount of crosslinks is low and adsorption of the polymer subchains to the interface is not accompanied by a high penalty in elasticity. Therefore, ULC

microgels form a pancake-like shape, with a homogeneous core and thin fuzzy shell. In other words, strong adsorption overcomes elasticity. Compression and interpenetration lead to a higher polymer fraction inside the microgels, making them stiffer since the elasticity of networks depends on the polymer fraction within the microgels[10,29]. Therefore, for further compression of the monolayer, an increase of the concentration of polymer at the interface becomes energetically unfavorable with respect to desorption and folding some of the polymer subchains of the microgels out of plane. As a consequence, the ULC microgels tend to restore their individual shape: the stronger the compression, the higher the protrusion.

ULC microgels also present differences with respect to other soft colloidal objects like block copolymer micelles, which irreversibly change their internal conformation upon adsorption at an interface, exposing hydrophilic and hydrophobic blocks to the polar and non-polar phases, respectively[48,49]. In contrast, the transition of ULC microgels from a homogeneous coverage to disordered microgel-like layer is fully reversible. Furthermore, the interfacial phase behavior of ULC microgels, that only show disordered arrangements, differs from colloids that can crystallize[50,51].

This can be explained since the confinement of the ULC microgels in two dimensions produces a heterogeneous stretching of their loosely crosslinked polymeric networks. This is a consequence of both the few crosslinks they have and the absence of a well-defined crosslinking profile: a more crosslinked core will preserve more the shape and limits the increase in size polydispersity as for regularly crosslinked microgels. The consequent increase in the size polydispersity of the ULC microgels suppresses the crystallization of the system. With increasing the compression of the monolayer of ULC microgels their size distribution becomes narrower, but still too broad to allow crystals to form[46]. Even assuming that as in 3D[13] a higher compression further decreases the polydispersity, microgels are jammed together, making any rearrangements impossible and the materials fail to crystallize.

The capability of ULC microgels to behave either like flexible, linear polymers or like colloids depending on dimensionality and compression makes them the perfect model system to explore common properties and differences between them and regular microgels and other polymer architectures as star, hyperbranched and linear macromolecules. Consequently, these materials may also be of great interest for applications as biomaterials. For example, since ULC microgels are more deformable than the regular crosslinked microgels[21], they are particularly suitable for the development of nanomaterials for selective adsorption in physiological solution[20], or as bio-sensors[52], as recent findings show that the softness of microgels improves their adsorption onto a solid substrate[53]. Furthermore, it has been proven that ultra-low crosslinked microgels are a perfect system to generate porous fibrin networks facilitating cell migration and growth[54]. The capability of this system to have a topography under adsorption that depends on the compression and the suppression of crystallization in two dimensions would be of interest in these fields. Here we have also shown that while ULC microgels have a behavior similar to linear polymers at the interface, their flow properties are more comparable with colloids. This might be of interest to realize polymer coatings with surface morphologies ranging from flat, uniform films, e.g., linear or star polymers[55], to randomly close-packed monolayers of microgels[56].

## Methods
**Synthesis**. Ultra-low crosslinked microgels were synthesized by precipitation polymerization according to ref. [21]. Briefly, 3.9606 g NIPAM and 0.1802 g SDS (sodium dodecyl sulfate) were dissolved in 495 mL filtered (0.2 μm regenerate cellulose membrane filter) double-distilled water. In contrast to ref. [21], SDS was used to have a better control on the size polydispersity of the particles[57]. The monomer solution was purged with nitrogen under stirring at 100 rpm and heated to 70 °C. Separately, 0.2108 g of KPS (potassium peroxydisulfate) in 5 mL filtered double-distilled water was degassed for 1 h. The polymerization was initiated by transferring the KPS solution using a nitrogen-washed syringe and needle into the monomer solution. The reaction was left to proceed for 4 h under constant nitrogen flow at 70 °C and 100 rpm. The resulting microgels were purified by threefold ultra-centrifugation at 50,000 rpm and subsequent redispersion in fresh water. During the synthesis many polymeric chains are trapped within the collapsed precursor particles but they are not linked to the polymeric network. These chains were then washed out of the particles during the purification process. Lyophilization was applied for storage.

The yield of the precipitation polymerization for ULC microgel formation is only ≈10%, while for regularly crosslinked microgels it is ≈90%. This means that many chains containing initiator fragments carrying charges are not incorporated into the ULC microgels. In other words, only a few of the formed chains become part of the polymeric network of the ULC microgels. Therefore, those microgels are expected to incorporate a negligible amount of charges that will not produce significant effects on the interaction potential between the ULC microgels.

The synthesis of regular 5 mol% crosslinked pNIPAM microgels was done by precipitation polymerization. The monomers NIPAM (5.4546 g), BIS (0.3398 g) and AMPH (N-(3-aminopropyl)acrylamide, 0.1474 g) were dissolved in 330 mL double-distilled water. Under stirring (270 rpm) the monomer solution was heated to 65 °C and purged with nitrogen. Separately, 0.2253 g V50 (2,2′-azobis(2-methylpropionamidine) dihydrochloride) and 0.0334 g of CTAB (cetyltrimethylammonium bromide) were each dissolved in 20 mL water in two separated vessels and degassed for 1 h. The surfactant was injected to the reaction vessel and stirred for additional 30 min to equilibrate. The polymerization was initiated by adding the V50 solution. For microgels in the range of size used for this work, the use of a particular initiator, e.g., V50 or KPS, has no effects on the phase behavior of the monolayer at the interface[14,34,35]. The reaction was carried out for 4 h at 65 °C under constant nitrogen flow and stirring. The obtained microgels were purified by threefold ultra-centrifugation at 30,000 rpm and subsequent redispersion in fresh, double-distilled water. Lyophilization was applied for storage.

**Small-angle neutron scattering**. Small-angle neutron scattering experiments were performed at the KWS-2 instrument operated by JCNS at the Heinz Maier-Leibnitz Zentrum (MLZ), Garching, Germany. The q-range of interest was covered using a wavelength for the neutron beam of $\lambda = 0.5$ and 1 nm and three sample-detector distances: 20, 8 and 2 m. The scattering vector is $q = 4\pi/\lambda \, \sin(\theta/2)$, with $\theta$ the scattering angle. The detector is a 2D-$^3$He tubes array with a pixel size of 0.75 cm. The relative error on the wavelength is $\Delta\lambda/\lambda = 10\%$. The data were corrected accounting for sample transmission and dark count ($B_4C$ used). The background, heavy water, has been subtracted from all data.

**Dynamic light scattering**. A laser with vacuum wavelength $\lambda_0 = 633$ nm was used to probe diluted suspensions of the different microgels in water, with refractive index $n(\lambda_0) = 1.33$. The temperature was changed between 10 and 50 °C, in steps of 2 °C, using a thermal bath filled with toluene to match the refractive index of water. The scattering vector $q = 4\pi n/\lambda_0 \sin(\theta/2)$ was changed by varying the scattering angle, $\theta$, between 30 and 130°, in steps of 5°.

**Capillary viscosimetry**. To obtain the viscosity, the average time of fall, $t$, of a constant volume of microgel suspension through a thin capillary of an Ubbelohde tube viscometer immersed in a water bath at a fixed temperature of $20.0 \pm 0.1$ °C was measured. The average fall times of the suspensions at different concentrations are linked to the kinematic viscosities, $v$, by a constant, $C$, that only depends on the geometry of the capillary: $v = Ct$; in our experiments, $C = 3.156 \times 10^{-9}$ m$^2$ s$^{-2}$. By knowing the sample density, which can be approximated with that of water ($\rho_{H_2O}$) due to the low concentration of microgels in all the measured suspensions, the viscosity of the microgel suspensions as a function of microgel concentration can be computed, $\eta = v\rho_{H_2O}$.

**Rheology**. Oscillatory rheology has been used to probe the flow properties of the suspensions. A Kinexus Pro Rheometer (Malvern Panalytical Ltd) with a cone-plate geometry (40 mm, 1.0°) was used. Before each frequency sweep, amplitude sweeps were performed to verify that the suspensions were in the linear viscoelastic regime at the frequencies $\omega = 0.1$, 1 and 10 rad s$^{-1}$ and in the range of $\gamma_{strain\%} = 0.5$–30%. At the end of the oscillatory experiments, a final amplitude sweep at $\omega = 1$ rad s$^{-1}$ was repeated to check that the system was still in the linear viscoelastic regime.

**Langmuir–Blodgett trough**. The mechanical properties of two-dimensional monolayers of the ultra-low crosslinked microgels, regular 5 mol% microgels and linear p(NIPAM) at the decane–water interface were probed using a Langmuir–Blodgett trough with customized dipping well (KSV NIMA, Biolin Scientific Oy). The same strategy to synchronize the continuous compression of the monolayer and the deposition of the microgels to a solid substrate, reported

in ref. [14], was used. Two parallel-moving barriers were closed ($v = 6$ mm min$^{-1}$) to increase the concentration of microgels at the interface by decreasing the available area. To probe the surface pressure, a platinum Wilhelmy plate parallel to the barriers attached to an electronical film balance was used. At the same time, a cleaned piece of silicon wafer was lifted up in between the barriers at an angle of ≈20° with respect to the interface. Thus, the monolayers were immobilized in a Langmuir–Blodgett-type deposition on silicon wafers and subsequently investigated ex situ by atomic force microscopy. Compression isotherms and depositions were conducted at 20.0 ± 0.5 °C.

**Atomic force microcopy.** AFM measurements were performed using a Dimension Icon with closed loop (Veeco Instruments Inc., software: Nanoscope 8.15, Bruker Corporation). The measurements of the microgels in the dry state, at the solid–air interface, were recorded in tapping mode with TESPA tips with a resonance frequency of 320 kHz, a nominal spring constant of 42 N m$^{-1}$ of the cantilever and a nominal tip radius of ≈8 nm (Bruker Corporation). Obtained AFM images were processed with the analysis software Gwyddion 2.48. The images were leveled to remove the tilt, the mean values were fixed to zero height and converted to grayscale.

**Analysis of atomic force micrographs.** The AFM micrographs of ultra-low crosslinked and regular 5 mol% crosslinked microgels were analyzed with a modified version of the custom-written Matlab script in ref. [14]. It is built around the Matlab version of the publicly available IDL particle tracking code by Crocker and Grier[58].

## Data availability

The data that support the findings of this study are available at https://hdl.handle.net/21.11102/0f3e60d7-9576-11e8-9b95-e41f1366df48

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

## Acknowledgements

A.S. thanks Alexander von Humboldt Foundation for financial support. The authors acknowledge the Deutsche Forschungsgemeinschaft (DFG) within the SFB 985 Functional Microgels and Microgel Systems and the International Helmholtz Research School of Biophysics and Soft Matter (IHRS BioSoft) for the financial support. L.I. and M.A.F.-R. acknowledge financial support from the Swiss National Science Foundation grants PP00P2_144646/1 and PP00P2_172913/1. I.I.P. acknowledges the Government of the Russian Federation within Act 211,Contract No. 02.A03.21.0011 for the financial support. This work is based upon experiments performed at the KWS-2 instrument operated by Jülich Centre for Neutron Science (JCNS) at the Heinz Maier-Leibnitz Zentrum (MLZ), Garching, Germany.

## Author contributions

A.S. and S.B. equally contributed to the paper. W.R. and A.S. designed the study. A.S., J.E.H. and M.B. performed scattering experiments. A.S. analyzed scattering experiments. M.A.F.-R., M.F.S. and S.B. performed AFM measurements. S.B. analyzed AFM data. A.S., S.B. and A.P.H.G. performed Rheology measurements. A.S. and A.P.H.G. analyzed Rheology measurements. M.B., A.P.H.G. and S.B. synthesized the samples. All authors took part in discussions of the results, completion and commented on the manuscript.

## Additional information

**Competing interests:** The authors declare no competing interests.

