## [Peer Review File · Nature Communications]

Reviewers' comments:

Reviewer #1 (Remarks to the Author):

The authors claim that poly(NIPAM) spheres with a very low crosslink density, which they refer to as ULC (ultra low crosslink) microgels, exhibit phase behavior and rheology similar to soft colloids in three dimensions, but display mechanical properties and phase behavior more similar to linear polymers when they are confined to two dimensions; e.g., at an oil-water interface.

I find this to be a fascinating study, and I believe the results will be of significant interest to the community. The broad idea of when a soft colloid begins to behave more as a polymer, and when concepts from colloids can be applied to macromolecules, is an important question in the soft matter and protein biophysics communities. There has been work on this question in three dimensions (for example, bulk rheology studies by D. Vlassopoulos and others on polymer stars and colloids with grafted polymer layers) that focus on the transition from "polymer behavior" to "colloid behavior." However, to my knowledge, there are no experimental studies to demonstrate that this transition is dependent upon the geometry or dimensionality of the system. Thus, I believe the work will have high impact and has the potential to influence thinking and future work in this field, and is appropriate for publication in Nature Communications.

The SANS, rheology, surface pressure data, and AFM images appear to be of good quality, with sufficient statistics for the SANS. The results do support the authors' conclusions; specifically, the surface pressure data (e.g., the comparison between the ULC, a more heavily crosslinked microgel bead, and linear p(NIPAM) shown in Figure 3) and the comparison of the phase behavior in 2 and 3 dimensions (e.g., the crystals of ULC shown in Figure 2 as compared to the lack of crystalline order in the AFM images of Figure 4) are particularly convincing.

I do not have any major criticisms of the paper. A minor comment is that, in the discussion of Figures 3 and 4, it may be good to briefly discuss the expected behavior of hard spheres for the surface pressure and for the two-dimensional phase behavior. They have included this in the discussion of Figure 2 in terms of the 3D phase behavior of hard spheres. I believe it would be helpful to the reader and may increase the impact of the results. Other than this, it is a very strong contribution and beautiful work, and I recommend publication in Nature Communications. (Surita Bhatia, Stony Brook University)

Reviewer #2 (Remarks to the Author):

"The colloid-to-polymer transition: Soft, ultra-low cross-linked microgels behave as colloids in three- and as polymers in two dimensions", A. Scotti et.al.

I think the results in this paper are interesting. The experiments are well performed and the interpretation is sound. However, I have a major concern, which is addressing the "why". I do not feel the authors provide the why behind their interesting observations. ULC microgels contain self-crosslinks. How many? How would this number compare to the number in regular pnipam-bis microgels with the smallest possible amount of bis? How are these cross-links distributed in ULC microgels? Are they more homogeneously distributed, as one might perhaps expect? If yes, then why would they exhibit more interpenetration than regular pnipam-bis microgels in two-dimensions? I am not sure the authors provide a fundamental explanation to what they observe, and I believe that they must do this before the paper is published. Why do ULC microgels behave the way they do compared to slightly cross-linked pnipam-bis microgels?

I also have other comments/questions:

* The "two-stepped course" referred to in page 3 (in reference to Fig. 3) should be discussed

more. Do they mean the initial decay and then the final one in the cyan data points in Fig. 3?

* What are the errors in Fig. 3? This is extremely important. The data in the inset are considered different in the discussion. However, this depends on the errors in the measurements? How many independent trials were done? What's the associated standard deviation?

* It is not obvious to see that the structure in Fig. 4(c) is "another hexagonal lattice with a smaller lattice constant". In addition to the ψ_6 , other less local measures should be provided.

* At the beginning of the discussion, the authors must explicitly say that the "properties are very similar to flexible polymers" within certain ζ range.

Reviewer #3 (Remarks to the Author):

Scotti and co-workers report on the behavior of ultra-low cross-linked microgels in bulk solution and at interfaces. Such microgels are very interesting model systems. In bulk solution they are found to behave colloid-like. However, at interfaces already at low shear forces their behavior is polymer-like and they tend to form films. Hence, this is really an ideal model system to study the transition from colloids to polymers. The manuscript is well written and presents excellent experimental data. I think this is a very nice work and it should be published after minor revision.

The following points should be revised:

-The interaction potential is also determined by the charges introduced by KPS during the polymerization. To what extent is this important? Please comment on this. Did you try to remove the SO_4 groups by acid treatment? This might be interesting since this would make the potential even softer.

- In fig. 1 the fuzzy sphere model is also used for the collapsed state of the particles. Wouldn't it be better to fit the 40°C data with a hard sphere model?

-Why did you use cationic "normal microgels" for comparison?

Wouldn't it be easier to make just simple NIPAM homopolymer microgels also with KPS as initiator?

some typos:

line 5 of the abstract: ...properties are those of

page 2, right column, line 2: ...a disordered fluid...

page 3: Isn't the Langmuir trough only the Langmuir trough?

I think only the vertical deposition technique is called Langmuir-Blodgett.

page 4, left column, line 15: please replace "shape" by "cross-section"

page 4, right column, line 9: ...independent of the

page 6, section "Rheology": You certainly mean "cone-plate", isn't it?

We are really thankful to the editor for considering our paper for publication and to the Reviewers for the useful comments that have improved the robustness and clarity of our work.

The actions we have done to address the comments are reported in *italic* in this letter. The changes in the main manuscript and in the Supplementary Information are marked in red.

I. Reviewer #1

The authors claim that poly(NIPAM) spheres with a very low crosslink density, which they refer to as ULC (ultra low crosslink) microgels, exhibit phase behavior and rheology similar to soft colloids in three dimensions, but display mechanical properties and phase behavior more similar to linear polymers when they are confined to two dimensions; e.g., at an oil-water interface.

I find this to be a fascinating study, and I believe the results will be of significant interest to the community. The broad idea of when a soft colloid begins to behave more as a polymer, and when concepts from colloids can be applied to macromolecules, is an important question in the soft matter and protein biophysics communities. There has been work on this question in three dimensions (for example, bulk rheology studies by D. Vlassopoulos and others on polymer stars and colloids with grafted polymer layers) that focus on the transition from "polymer behavior" to "colloid behavior." However, to my knowledge, there are no experimental studies to demonstrate that this transition is dependent upon the geometry or dimensionality of the system. Thus, I believe the work will have high impact and has the potential to influence thinking and future work in this field, and is appropriate for publication in Nature Communications.

The SANS, rheology, surface pressure data, and AFM images appear to be of good quality, with sufficient statistics for the SANS. The results do support the authors' conclusions; specifically, the surface pressure data (e.g., the comparison between the ULC, a more heavily crosslinked microgel bead, and linear p(NIPAM) shown in Figure 3) and the comparison of the phase behavior in 2 and 3 dimensions (e.g., the crystals of ULC shown in Figure 2 as compared to the lack of crystalline order in the AFM images of Figure 4) are particularly convincing.

Reply:

We warmly thank Reviewer #1 for the positive report about our study.

I do not have any major criticisms of the paper. A minor comment is that, in the discussion of Figures 3 and 4, it may be good to briefly discuss the expected behavior of hard spheres for the surface pressure and for the two-dimensional phase behavior. They have included this in the discussion of Figure 2 in terms of the 3D phase behavior of hard spheres. I believe it would be helpful to the reader and may increase the impact of the results. Other than this, it is a very strong contribution and beautiful work, and I recommend publication in Nature Communications. (Surita Bhatia, Stony Brook University)

Reply:

We agree with Reviewer #1 that the comparison with the behavior of hard spheres at interfaces is of interest and it will complete the discussion about ULC microgels in 2D. We now refer to this explicitly in the manuscript and report citation to work where the phase diagram of a 2D gas of hard spheres is discussed in detail^[1]. In particular, for an ideal gas of hard spheres, π is expected to be determined by the excluded area effects and the collision rate before to diverge once particles are in contact^[1]. Concerning the phase behavior of hard spheres, they undergo to liquid-to-crystal transition forming hexagonal lattice with local order^[1,2].

Actions:

1. In section “Mechanical response of monolayers”, we added a sentence in the first paragraph of page 4 left column, to elucidate the expected course of the compression isotherms of hard spheres. We added the reference to the work of Luding where a quantitative model is developed for a 2D-gas of hard spheres.
2. In section “Two dimensional phase behavior”, in the fifth paragraph of page 4 right column, we added a sentence stating that a 2D-gas of hard spheres is expected to undergo liquid-to-crystal transition. Two references are also added, where a quantitative description of the liquid-to-crystal transition is given.

II. Reviewer #2

“The colloid-to-polymer transition: Soft, ultra-low cross-linked microgels behave as colloids in three- and as polymers in two dimensions”, A. Scotti et.al.

I think the results in this paper are interesting. The experiments are well performed and the interpretation is sound. However, I have a major concern, which is addressing the “why”. I do not feel the authors provide the why behind their interesting observations.

Reply:

We appreciate the positive response of Reviewer #2 and in the following we will make it clearer the “why” behind our observations.

ULC microgels contain self-crosslinks. How many?

How would this number compare to the number in regular pnipam-bis microgels with the smallest possible amount of bis?

Reply:

It is very difficult to estimate the amount of crosslinks incorporated within microgels. It is also impossible to distinguish between crosslinks due to self-crosslinking of pNIPAM promoted by KPS and crosslinks induced by the BIS. However, the swelling behavior below and above the VPTT can be a measure of the softness of the polymeric network. Recently, Lopez and Richtering have collected literature data and found a strong correlation between the effective amount of crosslinker incorporated within a microgel, f , and the hydrodynamic radii below and above the VPTT^[3]. The ULC microgels used here were characterized using multi-angle dynamic light scattering to obtain their hydrodynamic radius, R_h , below and above the volume phase transition. We obtained $R_h = (134 \pm 1)$ nm at 20 °C and $R_h = (44.8 \pm 0.2)$ nm at 40 °C. Those values can be used in the relation $R_h(20\text{ °C})/R_h(40\text{ °C}) = 0.966 \cdot f^{-0.2}$ ^[3] to estimate f .

For the ultra-low crosslinked microgels used here, we obtain $f = 0.35\%$. When this value is compared to the data reported in literature and collected in Ref.^[3], we see that for regularly crosslinked microgels of comparable size to the one used here, synthesized with a wt% of crosslinker equal to 0.89 wt%^[4], f results to be equal to 1.32 mol%. Also for the other microgels reported in Ref.^[3], f always results in values larger than 0.35%. This confirms that the ULC microgels have the polymeric network that contains the lowest amount of crosslinker.

Actions:

- 1. In page 2, we added a paragraph at the end of section “Small-angle neutron scattering” where we discuss the swelling behavior and the fact that ULC microgels are softer than microgels of comparable size synthesized with little crosslinker agent.*
- 2. In Supplementary material we have added the discussion above in the section “Swelling ratio and crosslinking”.*

How are these cross-links distributed in ULC microgels? Are they more homogeneously distributed, as one might perhaps expect?

Reply:

As expected by Reviewer #2, the crosslinks are more uniformly distributed within ULC microgels. This is highlighted both in bulk by SANS and at the interface as shown by the height and phase images (Fig. S5(a) and (b)). We apologize to have not made this clear enough in the previous version and we changed the manuscripts as indicated below.

Actions:

1. *On page 2, left column, we rephrased the first and third paragraphs of the section “Small-angle neutron scattering” to clarify that from SANS it is evident that ULC micorgels have a larger region with constant polymer density and, therefore, crosslinks.*
2. *On page 4, left column, we expanded the first paragraph and on page 4, right column, we expanded the second and third paragraphs making it clearer that also in 2D the ULC microgels show a uniform polymer density. This differs from microgels synthesized with a little amount of crosslinker agent, which always show a more rigid and less extended core with higher polymer density. We added the reference to the work of Mourran and coworkers^[5].*

If yes, then why would they exhibit more interpenetration than regular pnipam-bis microgels in two-dimensions?

Reply:

We have probably not expressed ourselves in a clear manner in the original manuscript. In Fig. 4(e) the substrate is uniformly covered by ULC microgels which is a typical phenomenon observed once linear polymers are deposited on a substrate. Whether this is due to a more pronounced interpenetration with respect to regularly crosslinked microgels, or due to a higher deformation of the ULC microgels, cannot be determined from AFM. Also ellipsometry and neutron reflectometry do not give final answer to this point.

Anyhow, one can expect a stronger interpenetration of ULC microgels for two reasons: (i) the absence of a more crosslinked core; (ii) the fact that adsorption stretches the polymeric network of ULC microgels more than for regular microgels, as shown by their very flat height profile. This will increase the microgels mesh-size allowing for larger interpenetration. This could be addressed in further studies, maybe based on computer simulation, to elucidate the differences in the deformation of a 3D network once adsorbed on a 2D surface depending on the structure and topology of its network.

Actions:

1. *We rephrased the third paragraph of section “Two dimensional phase behavior” to make it clearer that either interpenetration or deformation, or both of them, can be responsible of the homogeneous coverage of the substrate shown in Fig. 4(e).*

2. *We added a citation to the work of Gumerov and coworkers where, with computer simulations, they show that arborescent polymers strongly deform at interfaces producing a uniform coverage of the substrate similar to the one observed here.*

I am not sure the authors provide a fundamental explanation to what they observe, and I believe that they must do this before the paper is published. Why do ULC microgels behave the way they do compared to slightly cross-linked pnipam-bis microgels?

Reply:

The different behavior of the ULC microgels with respect to the regularly crosslinked ones, once confined in two dimensions, is due to the differences between their polymeric networks. The poorly crosslinked network of the ULC microgels, together with the absence of a more crosslinked core that provides more resistance to deformations due to the confinement, lead the ULC microgels to spread radially and deform upon adsorption at the interface.

Since ULC microgels have a randomly crosslinked network with very low number of crosslinks it is reasonable to expect that every microgels posses a different topology. Once confined, the stretching of the different networks produces large variation in size with a consequent increase in size polydispersity. This suppresses crystallization. This effect is not so dramatic for crosslinked microgels with a significantly more crosslinked core that deform less and consequently do not show an increase in their size polydispersity in 2D.

We agree with the reviewer that, together with our observations, we did not provide a model to predict the different stretching of ULC microgels at the interface. We anyway hope that our work will stimulate theoreticians to develop models and methods to fully understand our observations. We would like to point out that developing a model that accounts for heterogenities of the polymeric network both in plane and in the z -direction will be extremely challenging. The difficulty is that microgels are three dimensional objects confined in quasi-2D environments and therefore they cannot be approximated as simple soft-disks.

Actions:

1. *In section “Two dimensional phase behavior”, page 5, the reason for the suppression of the crystallization is discussed in the second paragraph of the left column.*
2. *We added a third paragraph to the section “Two dimensional phase behavior”, in page 5, left column, where we discuss the reason for the increase of size polydispersity.*
3. *In page 6, left column, we added the second and third paragraphs to the discussion*

where we summarize the reasons for the unique concentration dependent topography observed for ULC microgels.

4. In page 6, left column, we added the fifth paragraph to the discussion where we summarize the reasons for the suppression of the crystallization for ULC monolayers.

I also have other comments/questions:

***The “two-stepped course” referred to in page 3 (in reference to Fig. 3) should be discussed more. Do they mean the initial decay and then the final one in the cyan data points in Fig. 3?**

Reply:

The Reviewer #2 is right, we have now made it clearer what we meant with “two-stepped course”. Similar discussion and description of the compression isotherms of microgels at interfaces can also be found in in Refs. [6,7,8].

Actions:

1. In the third paragraph of the right column on page 3, we added a more precise description of the “two-stepped course” with reference to Fig. 3 in the section “Mechanical response of monolayers”.

***What are the errors in Fig. 3? This is extremely important. The data in the inset are considered different in the discussion. However, this depends on the errors in the measurements? How many independent trials were done? What’s the associated standard deviation?**

Reply:

For clearness we have chosen to not include the error bars in the inset of Fig. 3. Two sources of errors are important in the Langmuir-Blodgett measurements: (i) the errors linked to the determination of the values of π and *Area*; (ii) the error related to the amount of microgels in the spreading solution that goes to the interface.

The nominal error related to a single measurement, as reported by the producer, is for the surface pressure, π , $0.1 \mu\text{N m}^{-1}$ and for the trough area 0.01 cm. Therefore, the error bars are within the symbols of the experimental points reported in Fig. 3 in the manuscript.

The shifts in x-axis mostly originate from the normalization of the trough area (*Area*) by the amount of microgels (*Mass*) added. The amount is calculated from the concentration and weight of the spreading solution. Differences in the concentration of the

Figure A1: Plot of π versus the trough area normalized by the trough area at the contact point, $Area/Area_{contact}$, of 5 mol% cross-linked microgels (cyan lines), ULC microgels (black lines) and linear pNIPAM (red lines) at $T = (20.0 \pm 0.5) ^\circ\text{C}$ in log-log scale including errors.

spreading solution are a source of error to be considered. More important is a partial loss of microgels into the sub-phase during addition, which might occur as microgels are soluble in the sub-phase.

Deviations in the surface pressure π between measurements of the same system stem from miss placement of the probe (Wilhelmy plate), impurities in the Langmuir trough, contamination of the spreading solution and vibrations of the setup. Experiments were conducted with utmost care to eliminate these errors.

To evaluate the impact of those errors on the reproducibility of a measurement, we conducted multiple measurements (with and without simultaneous deposition) as you can see e.g. in Fig. 3 (cyan curves). From the repeated measurements, at least six independent measurements, we have computed the standard deviation for the surface pressure, σ_π , and for the $Area/mass$, σ_x . The obtained values are: $\sigma_\pi \lesssim 0.3 \text{ mN m}^{-1}$ and $\sigma_x \lesssim 1 \%$. Fig. A1 shows the inset of Fig. 3 of the main manuscript showing the experimental errors. As can be seen the differences in the course of the curves are larger than the experimental errors. This confirms the robustness of our conclusions.

Actions:

1. In page 4, left column, second paragraph, we added a sentence regarding the errors and the robustness of our conclusion. We indicate that more information on the errors associated to a Langmuir-Blodgett trough measurements can be now found in the Supplementary Information.
2. We have included the discussion reported above and Fig. A1 within the Supplementary Information in a paragraph titled “Error propagation for the Langmuir-Blodgett measurements”.

***It is not obvious to see that the structure in Fig. 4(c) is “another hexagonal**

lattice with a smaller lattice constant". In addition to the ψ_6 , other less local measures should be provided.

Reply:

We do agree with Reviewer#2 that in the previous version of the manuscript it was not obvious that the two crystalline phases were different since we were reporting only the values of ψ_6 . To make this clearer Fig. A2 has to be considered.

Figure A2: Top left: compression isotherms as reported in Fig. 3 in the manuscript where the point corresponding to the AFM micrograph in (m) is marked, top right. Bottom: radial distribution functions corresponding to panels (b) and (c) of Fig. 4 in the main manuscript and probability distribution for nearest neighbour distance, d_{nn} , during the phase transition in image (m).

An AFM micrograph taken in position (m) of the compression curve of the 5 mol% crosslinked microgels is reported in the top right of Fig. A2. From this image it is clear that the previous crystalline structure, characterized by a larger lattice constant, is melting and at the same time crystalline grains of the new crystals are forming. These grains are characterized by a smaller lattice constant due to the compression of the fuzzy-shell.

To further prove this, the radial distribution functions corresponding to the different

crystalline phases, (b) and (c) are reported in the bottom line of Fig. A2. The two crystals are characterized by two different lattice constants, first peak in (b) and (c), corresponding to 590 and 150 nm. The dramatic decrease of the size is due to the compression of the microgels at the interface due to the movement of the barriers. To further highlight this, Fig. A2 reports the probability distribution for the nearest neighbour distance, $P(d_{nn})$, corresponding to the AFM micrograph at position (m). During the transition two distinct nearest neighbour distances emerge, one corresponding to the new lattice and one corresponding to the initial crystals that are melting. This is clearly demonstrated by the presence of a bimodal distribution in $P(d_{nn})$. The first peak, $d_{nn} = 180$ nm, corresponds to the new lattice while the second peak, $d_{nn} = 350$ nm, is related to the melting crystals.

The fact that we have two crystalline phases, characterized by two significantly different lattice constants and that the second lattice forms while the first one is melting, is consistent with studies of microgels at interfaces that observed the same behavior^[9,6,10].

Actions:

1. *In page 5, left column, we modified the first paragraph. Now we clearly refer to the presence of this solid-to-solid isostructural phase transition, via reference to the significant literature and the Supplementary Information.*
2. *We added the above discussion and Fig. A2 with the radial distribution functions to the Supplementary Information in the section “Solid-solid transition of the monolayer of regular microgels”.*

***At the beginning of the discussion, the authors must explicitly say that the “properties are very similar to flexible polymers” within certain ζ range.**

Reply:

We do agree with this observation and we have changed the manuscript according to this comment.

Actions:

1. *In page 2, first paragraph, we added a sentence.*
2. *In page 5, at the beginning of the discussion we added a sentence.*

III. Reviewer #3

Scotti and co-workers report on the behavior of ultra-low cross-linked microgels in bulk solution and at interfaces. Such microgels are very interesting model systems. In bulk solution they are found to behave colloid-like. However, at interfaces already at low shear forces their behavior is polymer-like

and they tend to form films. Hence, this is really an ideal model system to study the transition from colloids to polymers. The manuscript is well written and presents excellent experimental data. I think this is a very nice work and it should be published after minor revision.

Reply:

We really appreciate the positive comments of Reviewer #3 about our work.

The following points should be revised:

-The interaction potential is also determined by the charges introduced by KPS during the polymerization. To what extent is this important? Please comment on this. Did you try to remove the SO₄- groups by acid treatment? This might be interesting since this would make the potential even softer.

Reply:

The incorporation of charges within the ULC microgels is much lower than for regular ones, such that their effect on the interparticle potential can be neglected.

We highlight that during the synthesis many polymeric chains are trapped within the collapsed precursor particles but they are not linked to the polymeric network. These chains are then washed out of the particles during the purification process.

The yield of the precipitation polymerization for ULC microgel formation is only $\approx 10\%$, while for regularly crosslinked microgels is $\approx 90\%$. This means that many chains which contain initiator fragments carrying charges are not incorporated into the ULC microgels. In other words, only a few of the formed chains become part of the polymeric network of the ULC microgels. Therefore, those microgels are expected to incorporate a negligible amount of charges that will not produce a significant effect on the interaction potential between the ULC microgels.

Action:

1. In page 7, left column, we added an extract of the discussion above to the section "Synthesis".

- In fig. 1 the fuzzy sphere model is also used for the collapsed state of the particles. Wouldn't it be better to fit the 40°C data with a hard sphere model?

Reply:

The fit with the hard-sphere model leads to virtually the same results than the fit with the fuzzy-shell model. We choose to use the fuzzy sphere model to fit the form factor above the VPTT to not impose any *a priori* knowledge to our data and verify that the width of the fuzzy shell goes to its minimum during the fitting routine.

Actions:

1. *In the forth paragraph of page 2 right column, we added a sentence saying that the fits with the two models lead to virtually the same results and refer to the Supplementary Information.*
2. *In the Supplementary Information, we have included the fits and the radial profiles obtained fitting the data with both the fuzzy- and hard-sphere model in the section “Comparison between hard- and fuzzy-sphere model above the VPTT”.*

-Why did you use cationic “normal microgels” for comparison? Wouldn’t it be easier to make just simple NIPAM homopolymer microgels also with KPS as initiator?

Reply:

Literature shows that there is no difference both in the compression isotherms and in the phase behavior between cationic “normal microgels” and “normal microgels” synthesized with KPS or APS once they are confined at the interface^[6,7,11].

Here we chose to directly measure the behavior of microgels with a similar size compared to the ULC microgels. For this reason we chose to use the microgels synthesized with V50 since they had a radius comparable to the one of the ULC microgels, being sure that in this size range, the difference in charges with respect microgels obtained with KPS as initiator have no effect on both the compression isotherms and the phase behavior at the interface.

1. *In the first paragraph of page 7, right column, we added a sentence stating clearly that microgels synthesized with V50 or KPS have the same phase behavior once confined at the interface, independent to the charges of the initiator. We added three references showing this^[6,7,11].*

some typos:

line 5 of the abstract: ...properties are those of ...

page 2, right column, line 2: ...a disordered fluid...

page 3: Isn't the Langmuir trough only the Langmuir trough? I think only the vertical deposition technique is called Langmuir-Blodgett.

Reply:

We thank Reviewer#3 for this comment. The combination of surface pressure control and vertical or horizontal deposition is called Langmuir-Blodgett or Langmuir-Schaefer technique, respectively. A trough which allows the simultaneous compression and deposition is called Langmuir-Blodgett trough. As we performed compression isotherms and simultaneous deposition, we think that Langmuir-Blodgett trough is the appropriate term and we have not changed this in the manuscript.

Actions:

1. We made it clearer the procedure and therefore that we used a Langmuir-Blodgett trough in page 3 right column, first paragraph.

page 4, left column, line 15: please replace “shape” by “cross-section”

page 4, right column, line 9: ...independent of the ...

page 6, section “Rheology”: You certainly mean “cone-plate”, isn't it?

Reply:

We thank Reviewer#3 to have noticed these typos. We have changed in the main text.

Actions:

1. We corrected the typos in the manuscript.

References

- [1] S. Luding. Global equation of state of two-dimensional hard sphere systems. *Phys. Rev. E*, **63**, 042201 (2014).
- [2] S. Pronk & D. Frenkel Melting of polydisperse hard disks *Phys. Rev. E.*, **69**, 066123 (2004).
- [3] C. G. Lopez & W. Richtering. Does Flory-Rehner theory quantitatively describe the swelling of thermoresponsive microgels? *Soft Matter*, **13**, 8271–8280 (2017).
- [4] U. Gasser, J. S. Hyatt, J.-J. Lieter-Santos, E. S. Herman, L. A. Lyon & A. Fernandez-Nieves Form factor of pNIPAM microgels in overpacked states. *J. Chem. Phys.*, **141**, 034901 (2014).

- [5] A. Mourran, Y. Wu, R. A. Gumerov, A. A. Rudov, I. I. Potemkin, A. Pich & M. Möller. When Colloidal Particles Become Polymer Coils. *Langmuir*, **3**, 723-730 (2016)
- [6] M. Rey, M. A. Fernandez-Rodriguez, M. Steinacher, L. Scheidegger, K. Geisel, W. Richtering, T. M. Squires & L. Isa. Isostructural solid-solid phase transition in monolayers of soft core-shell particles at fluid interfaces: structure and mechanics. *Soft Matter* **12**, 3545–3557 (2016).
- [7] K. Geisel, L. Isa & W. Richtering. The Compressibility of pH-Sensitive Microgels at the Oil-Water Interface: Higher Charge Leads to Less Repulsion. *Angew. Chem. Int. Ed.*, **53**, 4905–4909 (2014).
- [8] F. Pinaud, K. Geisel, P. Massá, B. Catargi, L. Isa, W. Richtering, V. Ravaine & V. Schmitt. Adsorption of microgels at an oil-water interface: correlation between packing and 2D elasticity. *Soft Matter*, **10**, 6963–6974 (2014).
- [9] C. Picard, P. Garrigue, M.-C. Tatry, V. Lapeyre, S. Ravaine, V. Schmitt & V. Ravaine. Organization of Microgels at the Air-Water Interface under Compression: Role of Electrostatics and Cross-Linking Density. *Langmuir*, **33**, 7968–7981 (2017).
- [10] L. Scheidegger, M. A. Fernandez-Rodriguez, K. Geisel, M. Zanini, R. Elnathan, W. Richtering & L. Isa. Compression and deposition of microgel monolayers from fluid interfaces: particle size effects on interface microstructure and nanolithography. *Phys. Chem. Chem. Phys.*, **19**, 8671 (2017).
- [11] M. Rey, X. Hou, J. S. J. Tang & N. Vogel. Star-shaped Polymers through Simple Wavelength-Selective Free-Radical Photopolymerization. *Soft Matter*, **13**, 8717–8727 (2017).

REVIEWERS' COMMENTS:

Reviewer #1 (Remarks to the Author):

I believe the revised manuscript adequately addresses concerns raised in the first round of review. I would recommend publication in Nature Communications. (Surita Bhatia, Stony Brook University)

Reviewer #2 (Remarks to the Author):

Second report -- "The colloid-to-polymer transition: Soft, ultra-low cross-linked microgels behave as colloids in three- and as polymers in two dimensions", A. Scotti et.al.

I think the paper has improved with all additions. I nevertheless still have a few remarks that I believe should be clarified before publication:

1. In the reply to the referee comments

Page 2, in their reply – I urge the authors to more carefully discuss the behavior of hard spheres in 2d as compared to 3d, as they are qualitatively different. In 2d, there is an intermediate hexatic phase and the phase transitions (fluid-hexatic and hexatic-crystal) are mediated by topological defects. Talking about fluid-solid in 2d is thus insufficient.

Page 4, second reply. I do not understand the reply of the authors in any way. I strongly believe this has to be clarified. The authors argue that the ULCs exhibit more interpenetration than regular BIS microgels for two reasons. (i) The first one is that the ULCs lack a more crosslinked core. How can this contribute to the ULCs exhibiting more interpenetration? Since the BIS microgels have a more crosslinked core, the periphery will be less crosslinked compared to a microgel with a uniform crosslink distribution. One would then expect that the BIS microgels will interpenetrate more, as they have an incredibly fuzzy periphery. The authors must explain and justify how they conclude the opposite. (ii) The second reason is that adsorption stretches their polymeric network. Again, I think the authors must justify this. The argument I gave above implies the ULCs are already competing (in terms of who interpenetrates more) with a very fuzzy (with little crosslink) outskirts in the case of BIS microgels. How much will stretching increase the mesh size? Can the authors estimate this? Can they then compare the resulting mesh with that of the very uncrosslinked periphery of the BIS microgels? I am not convinced by the arguments given by the authors and feel they must justify their claims for them to be convincing.

I want to also add that there several answers to my comments, as well as to the other referees, which I have really liked. For example, their reply in page 3, and their second reply in page 10.

2. In the paper with corrections in red

Page 5, right after citing ref. 48. Could it be that being polydisperse changes the crystallization kinetics rather than hinder crystallization? Perhaps this additional possibility could be explored. There is recent work by Scotti on polydisperse and bidisperse suspensions that they can compare with and conclude whether this is a possibility that could be mentioned.

I think it is in the authors' best interest to clarify the points above in the paper. I trust they will do this and thus I do not need to see the paper. Once these comments are carefully considered, I recommend publication in Nature Communications.

Reviewer #3 (Remarks to the Author):

The authors have improved their manuscript according to the reviewer comments and have carefully addressed all issues which were mentioned in the reports.

This is really a nice and original work on the transition from colloid to polymer behavior.

Hence, I think the manuscript should be accepted in its present form.

I. Reviewer #1

I believe the revised manuscript adequately addresses concerns raised in the first round of review. I would recommend publication in Nature Communications. (Surita Bhatia, Stony Brook University)

Reply:

We thank Reviewer #1 for the very positive judgment on our work.

II. Reviewer #2

Second report – “The colloid-to-polymer transition: Soft, ultra-low cross-linked microgels behave as colloids in three- and as polymers in two dimensions”, A. Scotti et.al.

I think the paper has improved with all additions.

Reply:

We thank Reviewer #2 for the precious suggestions. We agree that the clearness and the impact of our paper have been improved due to the review process.

I nevertheless still have a few remarks that I believe should be clarified before publication: 1. In the reply to the referee comments ...

Page 2, in their reply – I urge the authors to more carefully discuss the behavior of hard spheres in 2d as compared to 3d, as they are qualitatively

different. In 2d, there is an intermediate hexatic phase and the phase transitions (fluid-hexatic and hexatic-crystal) are mediated by topological defects. Talking about fluid-solid in 2d is thus insufficient.

Reply:

We apologize for the imprecise treatment of the 2D phase behavior of hard disks and we thank Reviewer #2 to have highlighted this. Therefore, we changed the manuscript following the suggestion of Reviewer #2 and refer to a review of Gasser [*J. Phys.: Condens. Matter* **21** (2009) 203101] where the phase behavior of 2D hard disks is discussed.

Actions:

1. In page 5, lines 342-347, we added the sentence: “Similarly to the three dimensional case, a significant difference between colloids and flexible polymers at interfaces is that the former can have three different phases liquid, hexatic and crystalline. Those phases are characterized by two second-order transitions, meaning that no fluid-crystal or hexatic-crystal regimes can coexist [2].”

Page 4, second reply. I do not understand the reply of the authors in any way. I strongly believe this has to be clarified. The authors argue that the ULCs exhibit more interpenetration than regular BIS microgels for two reasons.

Reply:

We would like to highlight that in the paper we did not state that ULCs present more interpenetration. From the data and the techniques applied, particularly the AFM with reference to Fig. 4(e), it is not possible to establish whether interpenetration or compression/deformation is the dominant effect. We now state this more clearly on pages 4-5, lines 330-332.

Actions:

1. In pages 4-5, lines 330-332, we substituted the sentence “Thus, at low compressions, ULC microgels cover uniformly the interface as flexible polymers and not as colloids” with the sentence “Thus, at low compressions, ULC microgels cover uniformly the interface in contrast to colloids that always maintain their individual shape. Whether this is due to strong interpenetration, large deformation or both, cannot be deduced from Fig. 4(e).”

(i) The first one is that the ULCs lack a more crosslinked core. How can this contribute to the ULCs exhibiting more interpenetration? Since the BIS microgels have a more crosslinked core, the periphery will be less crosslinked compared to a microgel with a uniform crosslink distribution. One would

then expect that the BIS microgels will interpenetrate more, as they have an incredibly fuzzy periphery. The authors must explain and justify how they conclude the opposite.

Reply:

We agree with Reviewer #2 that we have not explained well what we meant. The key aspect is not the difference between the radial distribution of crosslinks but the absolute value of crosslinks within the microgels. In regular microgels the addition of crosslinker agent leads to a more crosslinked core and a fuzzy shell with fewer crosslinks with respect to the core. ULC have a much lower content of cross-links as compared to microgels prepared with a crosslinking agent. Consequently, the fuzzy shell of regular microgels is more crosslinked compared to the ULC microgels despite the fact that the ULC microgels have an almost uniform polymer distribution surrounded by a small fuzzy shell.

Figure A1: SANS intensities for 5 mol% crosslinked microgels used in Fig. 4 (orange circles), for microgels synthesized with 1 mol% BIS (green circles) and for the ULC microgels (light blue). The solid line are fits of the data using Eq. 1

As suggested by the reviewer, we estimated the mesh size of the polymeric network of the microgels using the high- q region of the SANS scattered intensity for the 5 mol% crosslinked microgels used in Fig. 4, for microgels synthesized with 1 mol% BIS and for the ULC microgels. Fig. A1 (which we added to the supplementary information) shows the SANS intensities, $I(q)$, and the fit of the data with:

$$I(q) = \frac{I_L(0)}{1 + \xi^2 q^2}, \quad (1)$$

where $I_L(0)$ is intensity at $q = 0$ and ξ is the mesh size of the network^[1,2]. From the fits we obtain that the ULC microgels have $\xi_{ULC} = 24 \pm 1$ nm while for the 1 and 5 mol% crosslinked microgels we obtain $\xi_{1 \text{ mol}\%} = 7.4 \pm 0.4$ nm and $\xi_{5 \text{ mol}\%} = 6.7 \pm 0.3$ nm, respectively. Those values for the mesh size are averaged over core and shell. Nevertheless they prove that, in average, ULC microgels present a network with larger meshes. We

would like to specify that as highlighted by SANS, ULC microgels possess a fuzzy external shell, as well. Both, the homogeneous region and the fuzzy shell of ULC microgels are expected to be less crosslinked than the fuzzy shell of regularly crosslinked microgels. This is reflected by the larger average mesh size. Therefore, it is reasonable to assume that ULC microgels have a more penetrable polymeric network.

Actions:

1. In page 2, lines 131-137, we added the sentence: “The average mesh size of the polymer network is estimated to be (24 ± 1) nm, which is significantly larger than for regular crosslinked microgels [23, 28]. This means that even if ULCs have a more homogeneous crosslinker distribution, the absolute number of crosslinks is lower than compared to regular crosslinked microgels”.
2. In the supplementary information, lines 94-124, we added the section “Determination of mesh size”, with Fig. A1 and the discussion reported above.

(ii) The second reason is that adsorption stretches their polymeric network. Again, I think the authors must justify this. The argument I gave above implies the ULCs are already competing (in terms of who interpenetrates more) with a very fuzzy (with little crosslink) outskirts in the case of BIS microgels. How much will stretching increase the mesh size? Can the authors estimate this? Can they then compare the resulting mesh with that of the very uncrosslinked periphery of the BIS microgels? I am not convinced by the arguments given by the authors and feel they must justify their claims for them to be convincing.

Reply:

A measure of the stretching of the polymeric network is given by the ratio between the radii of the microgels at the interface and in bulk, r_{2D} and r_{3D} . The radius of ULC microgels determined by the analysis of the AFM images is 308 nm while the hydrodynamic radius obtained by the analysis of DLS data is 134 nm. Therefore the stretching ratio is $r_{2D}/r_{3D} = 2.3$. The same ratio for the regular crosslinked microgels used in Fig. 4, is 1.9. This means that the polymeric networks of ULC microgels stretch more than the networks of regularly crosslinked microgels once adsorbed at the interface. As reported above, ULCs have a larger average mesh-size compared to regularly crosslinked microgels in bulk, therefore we can safely expect the mesh size of the ULC microgels to become even larger compared to the one of regularly crosslinked microgels once they are confined at the interface.

Actions:

1. We included those consideration in page 4, lines 300-309: “The ratio between the radii in two- and three-dimensions for ULC microgels is 2.3 while for the regularly

crosslinked microgels in Fig. 4(a) is 1.9. This ratio is a measure of the microgel stretching. The larger ratio of the ULC microgels suggests that their polymeric network is more stretched at the interface with respect to regularly crosslinked microgels. This observation and the fact that ULC microgels have a larger mesh size in bulk suggest that even at the interface the mesh size of ULC microgels is larger.”

I want to also add that there several answers to my comments, as well as to the other referees, which I have really liked. For example, their reply in page 3, and their second reply in page 10.

Reply:

We are pleased that our answers were satisfactory.

2. In the paper with corrections in red ...

Page 5, right after citing ref. 48. Could it be that being polydisperse changes the crystallization kinetics rather than hinder crystallization? Perhaps this additional possibility could be explored. There is recent work by Scotti on polydisperse and bidisperse suspensions that they can compare with and conclude whether this is a possibility that could be mentioned.

Reply:

Reviewer #2 is right, we cannot exclude that the increased polydispersity changes the crystallization kinetics. We changed the text accordingly.

Actions:

- 1. In page 5, we modified lines 363-368, which now read: “This can either significantly slow down the crystallization kinetics as observed in polydisperse suspensions of microgels in 3D [13], or suppress the crystallization since polydispersity has a value higher than the theoretical limit that hinders crystallization for hard disks in two dimensions [48].”*
- 2. We added an explicit reference to the study mentioned by Reviewer #2 in line 449, page 6, first paragraph in the right column.*

I think it is in the authors’ best interest to clarify the points above in the paper. I trust they will do this and thus I do not need to see the paper. Once these comments are carefully considered, I recommend publication in Nature Communications.

Reply:

We do agree and we hope that with the changes we made the manuscript can be considered suitable for the publication in Nature Communications.

III. Reviewer #3

The authors have improved their manuscript according to the reviewer comments and have carefully addressed all issues which were mentioned in the reports. This is really a nice and original work on the transition from colloid to polymer behavior. Hence, I think the manuscript should be accepted in its present form.

Reply:

We are happy to have addressed the comments of Reviewer #3 and that he is supporting the publication of our study on Nature Communications.

References

- [1] M. Stieger, W. Richtering, J. Pedersen, and P. Lindner, *J. Chem. Phys.* **120**, 6197 (2004).
- [2] A. Fernandez-Barbero, A. Fernandez-Nieves, I. Grillo, and E. Lopez-Cabarcos, *Phys. Rev. E* **66**, 051803 (2002).